Molecular barcode and morphological analysis of Smilax purhampuy Ruiz, Ecuador

Soledispa Pilar 1
Santos-Ordóñez Efrén gsantos@espol.edu.ec 2 3
Miranda Migdalia 4
Pacheco Ricardo 3
Gutiérrez Gaiten Yamilet Irene 5
Scull Ramón 5
1 Facultad de Ciencias Químicas. Ciudadela Universitaria “Salvador Allende”, Universidad de Guayaquil , Guayaquil , Ecuador
2 Facultad de Ciencias de la Vida, Campus Gustavo Galindo, ESPOL Polytechnic University, Escuela Superior Politécnica del Litoral, ESPOL , Guayaquil , Ecuador
3 Centro de Investigaciones Biotecnológicas del Ecuador, Campus Gustavo Galindo, ESPOL Polytechnic University, Escuela Superior Politécnica del Litoral, ESPOL , Guayaquil , Ecuador
4 Facultad de Ciencias Naturales y Matemáticas, ESPOL Polytechnic University, Escuela Superior Politécnica del Litoral, ESPOL , Guayaquil , Ecuador
5 Instituto de Farmacia y Alimentos, Universidad de La Habana , Ciudad Habana , Cuba
Sotelo-Mundo Rogerio
Electronic publication date: 2021 Mar 18
Publication date: 2021
Volume: 9
Electronic Location ID: e11028
Received 2020 Aug 5; Accepted 2021 Feb 8
Copyright: ©2021 Soledispa et al.
Copyright year: 2021
Copyright holder: Soledispa et al.
License: This is an open access article distributed under the terms of the Creative Commons Attribution License, which permits unrestricted use, distribution, reproduction and adaptation in any medium and for any purpose provided that it is properly attributed. For attribution, the original author(s), title, publication source (PeerJ) and either DOI or URL of the article must be cited.
License URL: https://creativecommons.org/licenses/by/4.0/

Keywords: atpF-atpH spacer, matK, rbcL, rpoB, rpoC1, psbK–psbI spacer, ITS2, Medicinal

Funding: The authors received no funding for this work.

==============================
Smilax plants are distributed in tropical, subtropical, and temperate regions in both hemispheres of the world. They are used extensively in traditional medicines in a number of countries. However, morphological and molecular barcodes analysis, which may assist in the taxonomic identification of species, are lacking in Ecuador. In order to evaluate the micromorphological characteristics of these plants, cross sections of Smilax purhampuy leaves were obtained manually. The rhizome powder, which is typically used in traditional medicines, was analyzed for micromorphological characteristics. All samples were clarified with 1% sodium hypochlorite. Tissues were colored with 1% safranin in water and were fixed with glycerinated gelatin. DNA was extracted from the leaves using a modified CTAB method for molecular barcode characterization and PCR was performed using primers to amplify the different loci including the plastid genome regions atpF-atpH spacer, matK gene, rbcL gene, rpoB gene, rpoC1 gene, psbK–psbI spacer, and trnH–psbA spacer; and the nuclear DNA sequence ITS2. A DNA sequence similarity search was performed using BLAST in the GenBank nr database and phylogenetic analysis was performed using the maximum likelihood method according to the best model identified by MEGAX using a bootstrap test with 1,000 replicates. Results showed that the micromorphological evaluation of a leaf cross section depicted a concave arrangement of the central vein, which was more pronounced in the lower section and had a slight protuberance. The micromorphological analysis of the rhizome powder allowed the visualization of a group of cells with variable sizes in the parenchyma and revealed thickened xylematic vessels associated with other elements of the vascular system. Specific amplicons were detected in DNA barcoding for all the barcodes tested except for the trnH–psbA spacer. BLAST analysis revealed that the Smilax species was predominant in all the samples for each barcode; therefore, the genus Smilax was confirmed through DNA barcode analysis. The barcode sequences psbK-psbI, atpF-atpH, and ITS2 had a better resolution at the species level in phylogenetic analysis than the other barcodes we tested.

Introduction

The genus Smilax (in the family Smilacaceae) consists of 310 species that are distributed in tropical, subtropical and temperate regions in both hemispheres of the world (Qi et al., 2013). According to Cameron & Fu (2006), Smilacaceae are taxonomically confused and belong to the cosmopolitan family of Liliales. Due to morphological analysis, the division of Smilacaceae includes at least seven genera and five sections within the large genus Smilax. Plants are dioecious, vine, herbaceous, or rarely, sub-shrubs or shrubs. Leaves are simple, and alternating with petioles that have tendrils; the primary venation are acrodomous. Thicker stems are rippled while aerial stems are generally aculeate (Martins et al., 2013b). In Ecuador, the genus is not well-recorded, although approximately ten species of this genus have been reported in the country according to Gaskin (1999).

Smilax is used a variety of ways in traditional medicine. For instance, in Brazil, Smilax longifolia Rich and Smilax syphilitica Humb & Bonpl. ex Wild are used as diuretics and in the treatment of venereal diseases (Breitbach et al., 2013) and Smilax quinquenervia Vell is used as a tonic for rheumatism and as an anti-syphilitic (Andreata, 1997). In Central America, several species of Smilax are used as diuretics, and for dermatological infections, gastrointestinal disorders, rheumatism, vaginitis, contraception, menstrual regulation, anemia, snake bites, and arthritis. In Ecuador, Smilax species are used for the elimination of cholesterol and triglycerides, the treatment of arthritis, intestinal, stomach and prostate inflammations, chronic gastritis, and cysts (Ferrufino & Gómez, 2004). Several pharmacological properties have been demonstrated, including glucose-lowering (Romo-Pérez, Escandón-Rivera & Andrade-Cetto, 2019), anti-hyperuricemic (Huang et al., 2019), anti-inflammatory and analgesic (Khana et al., 2019), diuretic (Pérez-Ramírez et al., 2016), and antioxidant (Fonseca et al., 2017) effects. Several chemical compounds have been identified in the genus, including polysaccharides (Zhang, Pan & Ran, 2019), steroidal saponins (Luo et al., 2018), and flavonoids (Wang et al., 2019), among others.

Smilax purhampuy is native to the Amazon and is distributed throughout Ecuador, Peru, Nicaragua, Colombia, Bolivia, Costa Rica, Venezuela, Honduras, and Brazil (Rivas et al., 2017). S. purhampuy is traditionally known for its healing and therapeutic properties. It has been used to treat cholesterol and triglycerides, chronic gastritis, cysts, arthritis, and intestinal, stomach, and prostate inflammations (JR Global del Perú, 2011). However, information on S. purhampuy is limited.

Morphological and molecular barcode analysis are lacking for S. purhampuy in Ecuador, despite its medicinal use. Phylogenetic analysis of the Smilax genus includes microsatellites (Martins et al., 2013a; Ru et al., 2017; Qi et al., 2017) which are also informative tools for genetic diversity and gene flow studies. Other methods for the phylogenetic analysis of plants includes DNA barcodes, which may be used as a complementary tool in the taxonomic identification of the species; for instance, the plastid genome regions atpF-atpH spacer, matK gene, rbcL gene, rpoB gene, rpoC1 gene, psbK–psbI spacer, and trnH–psbA  spacer have been tested as universal plant barcodes (CBOL Plant Working Group, 2009). The chloroplast genes rbcL and matK, are recommended to characterize land plants as a 2-locus combination (CBOL Plant Working Group, 2009).

The first reported use of the DNA barcode in Smilax species included the rDNA ITS sequence (Cameron & Fu, 2006) inferring phylogenetic relationships that elucidated the evolutionary and biogeographic history of the genera from the Smilacaceae family.

Sulistyaningsih et al. (2018) used the DNA barcode rbcL for phylogenetic analysis of Smilax spp. in Java, Indonesia. Qi et al. (2013) used the DNA barcodes ITS, matK and the rpl16 intron in Smilacaceae indicating that the phylogenetic relationships largely contradicted the traditional morphological classification of the family. Wang et al. (2014) used the DNA barcode psbA-trnH to distinguish Smilax glabra from its related species, and Kritpetcharat et al. (2011) used the trnH-psbA spacer barcode for Smilax china and S. glabra indicating that measuring the genetic distance may be used to discriminate between the two species. In a broader study which includes four species of Smilax and other trees, Liu et al. (2015) used the DNA barcodes rbcL, matK, ITS, ITS2, and trnH-psbA to analyze the diversity and species resolution, concluding that the combination of the loci rbcl + ITS2 is an effective tool for documenting plant diversity in the Dinghushan National Nature Reserve in China. Other loci in medicinal plants have been proposed for their characterization, including the nuclear sequence ITS2 (Zhang et al., 2016). Furthermore, DNA barcodes could be used to distinguish adulterated drugs (Kumari & Kotecha, 2016). Therefore, DNA barcode analysis should be performed for Smilax species to verify the results of taxonomic and morphological studies. We investigated the micromorphological and molecular barcode characterization of S. purhampuy Ruiz collected in Ecuador and found that the barcodes psbK-psbI, atpF-atpH, and ITS2 could be used in Smilax plants for a better resolution at the species level.

Materials and Methods

Study area

The climate of the study area is rainy megathermal with an average monthly temperature between 22 °C and 26 °C and average rainfall between 2,000 to 3,000 mm per year. The study area is a tropical humid forest.

Collection of plant material

Plant material was collected from three specimens of S. purhampuy Ruiz in the Francisco de Orellana Province in Ecuador (coordinates 1°10′03.7″S 76°56′30.9″W) in March and April of 2019. The samples were collected from shaded-exposed plants. Branches containing leaves, fruits, and the rhizome were transferred to the GUAY herbarium of the Faculty of Natural Sciences of the University of Guayaquil for taxonomic characterization. Samples were identified as S. purhampuy Ruiz (voucher number 13,117).

Plant material preparation

Leaves and rhizomes were washed with water. The leaves used in the micromorphological study were analyzed and stored at −80 °C for DNA extraction. The rhizomes were dried in a Mettler Toledo stove at 40 °C and then samples of the rhizomes were crushed with a manual knife mill and stored in amber glass jars for analysis.

Micromorphological analysis

Leaf samples were taken from the middle of the lamina for the evaluation of their micromorphological characteristics. The mid-rib was cut transversely according to the manual method (Miranda & Cuéllar, 2000). The maximum sample width was 1 cm including the mid-rib. Transversal cuts of fresh leaves were hydrated and clarified with 1% sodium hypochlorite. Tissues were colored with 1% safranin in water and fixed with glycerinated gelatin according to the method by Gattuso & Gattuso (1999). The powder obtained from the rhizomes was hydrated, clarified with 1% sodium hypochlorite and colored with 1% safranin in water and fixed with glycerinated gelatin (Gattuso & Gattuso, 1999; Miranda & Cuéllar, 2000). We performed a histochemical reaction with Lugol reagent to detect starch in the powdered drug obtained from the rhizomes (Gattuso & Gattuso, 1999). Morphological analysis was performed using a NOVEL light microscope at 10x magnification, attached to an HDCE-50B digital camera, model 146 HDCE-50B.

DNA extraction and PCR

Leaf samples from the three S. purhampuy Ruiz plants (with codes CIBE-010, CIBE-011, CIBE-012) were ground with MM400 (Retsch, Haan, Germany) and liquid nitrogen and were stored at −80 °C. DNA extraction was performed for each Smilax plant independently. A modified CTAB protocol was used for total DNA extraction according to Pacheco Coello et al. (2017). The master mix GoTaq® 2x (Cat# M7123, Promega) was used for PCR analysis according to the manufacturer’s instructions using 0.5 µM for each primer according to the barcode used (Table S1) in a 50 µL PCR reaction. The conditions for the PCR were: 95 °C for 3 min for initial denaturation; 35 cycles of 95 °C for 30 s, 50 °C/56 °C/60 °C (depending of the barcode, Table S1) for 60 s, 72 °C for 60 s; and a final extension of 72 °C for 10 min. Amplicons were detected by sampling 5 µL in agarose gel (1.5%) electrophoresis. The remaining 45 µL was purified and sequenced commercially (Macrogen, Rockville, MD, USA). At least two technical replicates were sequenced and a consensus was generated for each biological replicate.

Bioinformatic analysis

Sequences were processed using MEGAX (Stecher, Tamura & Kumar, 2020). Technical replicates were aligned with MUSCLE and a consensus sequence was generated for each barcode. Consensus sequences were analyzed by BLAST (Zhang et al., 2000) in the GenBank non-redundant nucleotide database (nr). The nr database included the accessions of the complete plastid genomes of Smilax spp. representing three species and also accessions containing sequences of single locus, indicating that the results were dependent on the sequence availability in the database at the time of the analysis (28th June 2020). Accessions were selected for phylogenetic analysis based on BLAST analysis. For each barcode, the accessions and samples sequences were aligned using MUSCLE and the recommended model from MEGAX was used. The aligned sequences were trimmed at the ends to allow for all sequences to maintain the same range. The maximum likelihood method was performed according to the best model found by MEGAX using bootstrap test (1,000 replicates).

Results

Morphological analysis

The micromorphological evaluation of a cross section of the leaf sample (Fig. 1A) showed a concave arrangement of the central midrib, which was more pronounced in the lower part with a slight protuberance. The mesophyll showed a well-defined adaxial and abaxial uniseriate epidermis with a fine cuticle on the lateral sides of the central vein. Below the adaxial epidermis, we observed a palisade parenchyma forming two or three continuous layers of elongated cells. The spongy parenchyma exhibited cells of variable size that bordered the abaxial epidermis. An enlargement of the central vein (Fig. 1B) showed the fundamental parenchyma, which was formed by many isometric cells. The sclerenchyma tissue was characterized by lignified thickened walled cells surrounding a well-defined vascular system (xylem and phloem) near the middle of the central vein, which harbored six vascular bundles of variable size.

Figure 1 Microscopic characteristics of the leaf from Smilax purhampuy Ruiz.

Transversal section of the central vein of the leaf (A, B). AbE, abaxial epidermis; AdE, adaxial epidermis; FP, fundamental parenchyma; Me, mesophyll; PP, palisade parenchyma; SP, spongy parenchyma; ST, sclerenchyma tissue; Vs, vascular system.

Micromorphological analysis of the drug obtained from the rhizome (Fig. 2) allowed the visualization of a group of cells of the parenchyma of variable size (Fig. 2A) and revealed thickened xylematic vessels associated with other elements of the vascular system (Fig. 2B). Elongated, fusiform, and pointed structures were also visualized, which corresponded to fibers and may suggest a type of filiform sclerides (Fig. 2C). We observed xylematic thickening vessels with holes in another sample of the powder drug (Fig. 2D). Numerous starch granules of variable size were observed showing a blackish coloration with the Lugol reagent (Fig. 2E).

Figure 2 Microscopic characteristics of the powder drug from Smilax purhampuy Ruiz rhizome.

(A) Parenchyma cells. (B) Xylematic veseels and other elements of the vascular system. (C) Fibers (filiform sclerides). (D) Xylematic veseels, (E) starch granules.

Molecular barcodes for Smilax purhampuy Ruiz plants

Specific PCR amplification was detected for all barcodes except for trnH-psbA (data not shown). BLAST analysis was performed for each barcode sequence (Data S1 and S2) and the best hit for all samples for each barcode indicated the Smilax species (Table 1). BLAST analysis indicated the presence of Smilax spp. using the sequences available at the nr database, including plastid genomes and single locus sequence. The best hits in BLASTn for the different species included: S. nipponica for psbK-psbI (96.57%, 96.56%, and 93.83% of identity for the three biological replicates, respectively); S. nipponica and S. china for rpoB (99.73% for the three biological replicates); S. nipponica, S. china, and S. aspera for rpoC1 (100%); S. sieboldii f. inermis for atpF-atpH (90.66%, 90.69%, and 96.47%), S. fluminensis (99.65%, 100%), S. bona-nox (99.88%) and S. laurifolia (99.88%) for matK; S. aspera (99.82%, 99.82%) and S. laurifolia (99.82%) for rbcL; and S. excelsa (80.49%, 80.05%) for ITS2 (only two biological replicates were sequenced successfully for ITS2).

Table 1 Blastn analysis for seven different barcodes of Smilax purhampuy Ruiz plants (CIBE-010, CIBE-011, CIBE-012).

Results were ranked for the first three with the highest percentage of identity. Species with the best results are bold for each barcode.

Barcode	Code	Blastn rank	
		1	2	3	
		Organism	Accesion	% identity	Organism	Accesion	% identity	Organism	Accesion	% identity	
psbK-psbI	CIBE-010	Smilax nipponica	MT261170.1a	96.57%	Smilax china	MT261168.1a	95.91%	Smilax glycophylla	MT261169.1a	92.69%	
	CIBE-011	Smilax nipponica	MT261170.1a	96.56%	Smilax china	MT261168.1a	95.91%	Smilax glycophylla	MT261169.1a	92.67%	
	CIBE-012	Smilax nipponica	MT261170.1a	93.83%	Smilax china	MT261168.1a	93.20%	Smilax glycophylla	MT261169.1a	89.32%	
rpoB	CIBE-010	Smilax nipponica	MT261170.1a	99.73%	Smilax glycophylla	MT261169.1a	99.73%	Smilax china	MT261168.1a	99.47%	
	CIBE-011	Smilax nipponica	MT261170.1a	99.73%	Smilax glycophylla	MT261169.1a	99.73%	Smilax china	MT261168.1a	99.46%	
	CIBE-012	Smilax nipponica	MT261170.1a	99.73%	Smilax glycophylla	MT261169.1a	99.73%	Smilax china	MT261168.1a	99.46%	
rpoC1	CIBE-010	Smilax nipponica	MT261170.1a	100.00%	Smilax china	MT261168.1a	100.00%	Smilax aspera	EU531650.1	100.00%	
	CIBE-011	Smilax nipponica	MT261170.1a	100.00%	Smilax china	MT261168.1a	100.00%	Smilax aspera	EU531650.1	100.00%	
	CIBE-012	Smilax nipponica	MT261170.1a	100.00%	Smilax china	MT261168.1a	100.00%	Smilax aspera	EU531650.1	100.00%	
atpF-atpH	CIBE-010	Smilax sieboldiif.inermis	JN417282.1	90.66%	Smilax sieboldii	JN417281.1	90.64%	Hemidesmus indicus	NC_047471.1	89.84%	
	CIBE-011	Smilax sieboldiif.inermis	JN417282.1	90.69%	Smilax sieboldii	JN417281.1	90.68%	Smilax glycophylla	MT261169.1a	89.03%	
	CIBE-012	Smilax sieboldiif.inermis	JN417282.1	96.47%	Smilax sieboldii	JN417281.1	96.31%	Smilax china	MT261168.1a	94.69%	
matK	CIBE-010	Smilax fluminensis	JF461414.1	99.65%	Smilax coriacea	KJ719950.1	98.72%	Smilax havanensis	KF782873.1	98.72%	
	CIBE-011	Smilax fluminensis	JF461414.1	100.00%	Smilax coriacea	KJ719950.1	99.05%	Smilax havanensis	KF782873.1	99.05%	
	CIBE-012	Smilax bona-nox	KC511353.1	99.88%	Smilax laurifolia	JF461393.1	99.88%	Smilax ligneoriparia	KX432989.1	99.76%	
rbcL	CIBE-010	Smilax aspera	KX394660.1	99.82%	Smilax aspera	KX394659.1	99.82%	Smilax aspera	KM609079.1	99.82%	
	CIBE-011	Smilax aspera	KX394660.1	99.82%	Smilax aspera	KX394659.1	99.82%	Smilax aspera	KM609079.1	99.82%	
	CIBE-012	Smilax laurifolia	JF944386.1	99.82%	Smilax china	MT261168.1a	99.45%	Smilax gaudichaudiana	KX394669.1	99.45%	
ITS2	CIBE-010	Smilax excelsa	JF461354.1	80.49%	Smilax aspera	KJ719926.1	80.24%	Smilax aspera	KJ719924.1	80.24%	
	CIBE-011	Smilax excelsa	JF461354.1	80.05%	Smilax aspera	KJ719926.1	79.81%	Smilax aspera	KJ719924.1	79.81	
Notes.

a Accession numbers refer to plastid complete genome.

Figure 3 Phylogenetic tree of the psbK-psbI spacer with accessions from the genus Smilax and different genera selected from the blastn results.

Three species from the genus Paris was used as outgroup. Maximum Likelihood method based on Tamura 3-parameter model. Boostrap test with 1,000 replicates was performed. The tree with the highest log likelihood (−1728.93) is shown. The percentage of trees in which the associated taxa clustered together is shown next to the branches. Initial tree(s) for the heuristic search were obtained automatically by applying Neighbor-Join and BioNJ algorithms to a matrix of pairwise distances estimated using the Tamura 3 parameter model, and then selecting the topology with superior log likelihood value. A discrete Gamma distribution was used to model evolutionary rate differences among sites (five categories (+G, parameter = 1.6020)). The tree is drawn to scale, with branch lengths measured in the number of substitutions per site. This analysis involved 26 nucleotide sequences. There were a total of 368 positions in the final dataset. Evolutionary analyses were conducted in MEGA X (Kumar et al., 2018; Stecher, Tamura & Kumar, 2020). Blue arrows indicate Smilax purhampuy Ruiz from Ecuador.

We determined the best models for phylogenetic analysis were: T92+G+I (rbcL), T92+G (matK, ITS2, psbK-psbI, atpF-atpH), T92 (rpoB), and JC (rpoC1) after alignment of the barcode sequences between the S. purhampuy from this study and different accessions including other genera. Phylogenetic analysis revealed that for psbK-psbI, the S. purhampuy sequences shared a clade (99 bootstrap) with different Smilax species, including S. china, S. nipponica, and S. glycophylla; and with two accessions from another genus, including Hemidesmus indicus (Fig. 3). However, the S. purhampuy sequences CIBE-010 and CIBE-011 are grouped in a subclade (bootstrap 96), while S. purhampuy CIBE-012 shared a clade with both S. purhampuy (bootstrap 30). The other major clades in the phylogenetic tree corresponded to other genera. For the rpoB the phylogenetic tree revealed that the S. purhampuy samples were in a clade (98 bootstrap) with other Smilax species; while other genera including Brahea spp., Philesia magellanica, Tricyrtis macropoda, Fritillaria spp., and Lilium spp. were in other clades (Fig. S1). The S. purhampuy samples for the rpoC1 were in a clade with different species of Smilax, including S. aspera (accession number EU531650), S. nipponica (MT261170), S. china (MT261168), and S. herbacea (HQ594138, HQ594139, Fig. S1). The atpF-atpH phylogenetic analysis revealed that several Smilax species were in the same clade. However, the S. purhampuy samples from this study are in a subclade (bootstrap 99). For the matK and rbcL barcodes, the same pattern was observed where a clade sharing different Smilax species was encountered (Fig. S1), although a subclade was formed with the two S. purhampuy samples (CIBE-010, CIBE-011) and S. fluminensis; while for the S. purhampuy CIBE-012 sample, a branch was shared with S. aspera for matK. Furthermore, in the rbcL phylogenetic tree, a subclade was formed for S. purhampuy samples CIBE-010 and CIBE-011 with other species including S. aspera, while for the S. purhampuy CIBE-012 a clade was shared with S. domingensis and S. lauriflora. The ITS2 phylogenetic tree (Fig. 4) revealed that the two S. purhampuy samples (CIBE-010 and CIBE-011) were in a different clade (bootstrap 99) apart from the other Smilax species, including S. aspera, S. stans, S. menispermoidea, S. trachypoda, S. aberrans, S. retroflexa, S. excelsa, S. lunglingensis, S. hispida, S. japonica, S. china, and S. pumila.

Figure 4 Phylogenetic tree of the ITS2 with accessions from the genus Smilax and different genera selected from the blastn results.

Three species from the genus Tulipa was used as outgroup. Maximum Likelihood method based on Tamura 3-parameter model. The tree with the highest log likelihood (−1405.71) is shown. The percentage of trees in which the associated taxa clustered together is shown next to the branches. Initial tree(s) for the heuristic search were obtained automatically by applying Neighbor-Join and BioNJ algorithms to a matrix of pairwise distances estimated using the Tamura 3 parameter model, and then selecting the topology with superior log likelihood value. A discrete Gamma distribution was used to model evolutionary rate differences among sites (five categories (+G, parameter = 0.8022)). The tree is drawn to scale, with branch lengths measured in the number of substitutions per site. This analysis involved 30 nucleotide sequences. There were a total of 436 positions in the final dataset. Evolutionary analyses were conducted in MEGA X (Kumar et al., 2018; Stecher, Tamura & Kumar, 2020). Blue arrows indicate Smilax purhampuy Ruiz from Ecuador.

Discussion

Microscopic analysis

We made a detailed assessment of the herbal drugs and used microscopy to identify them based on their known histological characteristics (Shailesh et al., 2015). Micromorphological studies are essential for the quality control of plant-derived drugs, since significant details are used to correctly identify the plant and possible adulterants.

The cross section of the S. purhampuy leaf indicated that the abaxial and adaxial epidermis are uniseriate, with the existence of an easily perceptible cuticle. Morphoanatomic studies performed on the leaves of various Smilax species (S. brasiliensis, S. campestris, S. cisoides, S. fluminensis, S. goyazana, S. oblongifolia, and S. rufescens) revealed a non-stratified epidermis with thick cuticle (Martins et al., 2013b). These results correspond to those of previous studies and may be a distinctive anatomical characteristic of the genus.

In most plants, the leaf mesophyll harbors palisade and spongy tissue, which differ in location, cell morphology, and function. In plants with a dorsiventral mesophyll, the palisade tissue is located on the adaxial side and the spongy tissue on the abaxial side; this distribution makes a greater contribution to the photosynthesis process (Yahia et al., 2019). Studies by various researchers have shown that the mesophyll could vary from one leaf to another of the same individual or of different individuals, depending on light intensity and salt concentration (Gapińska & Glińska, 2014). In S. purhampuy a dorsiventral mesophyll was observed where the palisade parenchyma is toward the adaxial side and the spongy parenchyma was located toward the abaxial side, which supports the results of Martins & Appezzato-da Glória (2006) for S. polyantha and Martins et al. (2013b) for S. brasiliensis, S. campestris, S. cissoids, S. goyazana, S. oblongifolia and S. rufescens. The presence of six vascular bundles of variable size at the level of the central vein of the leaf was unique and differed from other Smilax species (S. brasiliensis, S. cissoides, and S. fluminensis), which present three vascular bundles (Martins et al., 2013b).

The presence of parenchyma tissue, xylematic vessels, fibers and starch granules are reported in S. domingensis from Cuba and Guatemala (Cáceres et al., 2012; González et al., 2017), which was supported by the present study. The difference lies within the morphology and arrangement of these structures. For example, thickened scalariformly xilematic vessels and fibrotraqueids (fibers) were detected in S. domingensis, while thickened xilematic vessel with holes and filiform sclerides (fibers) were observed in the studied species.

Molecular barcode sequences

Genetic analysis has proven to be an important tool in the standardization of medicinal plants. The genotypic characterization of plant species is important since most plants may show considerable variation in morphology, although they belong to the same genus and species. DNA analysis is useful for the identification of cells, individuals, or species and could help distinguish genuine from adulterated drugs (Kumari & Kotecha, 2016). Different methods may be applied for genotyping in plants. Microsatellite (Martins et al., 2013a; Ru et al., 2017; Qi et al., 2017) and DNA barcodes (Cameron & Fu, 2006; Qi et al., 2013) have been used for genotyping Smilax species and have shown phylogenetic relationships between different Smilax species. Additionally, Sulistyaningsih et al. (2018) used the DNA barcode rbcL for phylogenetic analysis of Smilax spp. in Java, Indonesia, concluding that rbcL could be used to identify at the genus but not at species level.

The analysis of DNA barcodes could be used as a complementary analysis for the identification of plants species, especially when Smilax species show considerable phenotypic variations within populations (Cameron & Fu, 2006). The recommended barcodes for species identification are 2-locus rbcL and matK (CBOL Plant Working Group, 2009). Generally, the BLASTn analysis relies in the presence of those species in the GenBank for species identification; consequently, the results presented in the BLAST analysis and in the phylogenetic trees depended on the sequences available in the nr database in the GenBank. Therefore, a complete analysis using different species of Smilax should be performed in the future for all the DNA barcodes tested.

We analyzed the three samples taxonomically identified as S. purhampuy Ruiz (CIBE-010, CIBE-011, CIBE-012) which were more similar than other Smilax species, including S. nipponica, S. glycophylla, S. herbacea, S. china, S. sieboldin, S. aspera, S. stans, S. menispermoidea, S. trachypoda, S. aberrans, and S. pumila. These results were also observed in the phylogenetic trees for psbK-psbI spacer, atpF-atpH spacer, and ITS2. However, few accessions were encountered in the GenBank for the DNA barcodes psbK-psbI spacer and atpF-atpH spacer. There should be additional study of the different Smilax species for these barcodes; however, other DNA barcodes may accurately identify the genus level. The ITS2 revealed a low percentage of identification (80.49%) with BLASTn, suggesting that species differentiation could be detected using the ITS2. The results from other studies have indicated a better resolution for species identification using the ITS2 in medicinal plants (Techen et al., 2014; Zhang et al., 2016; Bustamante et al., 2019; Sarmiento-Tomalá et al., 2020).

Our results determined that the rpoC1 sequence was not accurate at the species level, and that matK could not be used to discriminate between S. purhampuy and S. fluminensis. Furthermore, the rbcL barcode could not be used for species differentiation in the Smilax genus, as a low bootstrap value was observed in the different clades formed. The psbK-psbI, atpF-atpH, and ITS2 had a better resolution at the species level for S. purhampuy. Future research should include the sequencing of selected barcodes (rbcL, matK, psbK-psbI spacer, atpF-atpH spacer, and ITS2) for different species of the Smilax genus found in Ecuador with biological replicates. Further studies should establish a reliable DNA barcode analysis and test different 2-locus combinations to determine which barcode should be used for species identification in the Smilax genus.

Conclusions

We determined the morphological characteristics and conducted molecular barcode analysis on S. purhampuy Ruiz plants collected in Ecuador. The micromorphological characteristics of the leaves and rhizomes were described for the first time, which constitutes a novel contribution to the botanical characterization of the species. The taxonomic classification of Smilax was confirmed by the molecular barcodes used, including psbK-psbI, rpoB, rpoC, atpF-atpH, matK, rbcL, and ITS2. Furthermore, the barcodes sequences psbK-psbI, atpF-atpH, and ITS2 indicated a better resolution at the species level than the other barcodes tested in this study. These barcodes (psbK-psbI, atpF-atpH, and ITS2) could be used to identify other species in the genus Smilax. However, further molecular barcode analysis should be performed on Smilax spp. from Ecuador to determine its diversity and to complete its taxonomic classification. Furthermore, the medicinal properties of the Smilax plants used in this study should be studied in greater detail.

Supplemental Information

Figure S1 Phylogenetic analysis of molecular barcode sequences

Click here for additional data file.

Table S1 Primers used for amplification of the barcodes psbA-trnH spacer, psbK-psbI spacer, rpoB, rpoC1, atpF-atpH spacer, rbc L, mat K, and ITS2

Click here for additional data file.

Data S1 Barcode DNA sequences of Smilax purhampuy Ruiz

Click here for additional data file.

Data S2 Accession numbers of Smilax purhampuy Ruiz (GUAY 13117) barcode sequences at GenBank

Click here for additional data file.

Identification of samples by the GUAY herbarium of the Faculty of Natural Sciences of the Guayaquil University is acknowledged.

Additional Information and Declarations

Competing Interests

Author Contributions

Field Study Permissions

Data Availability

The authors declare there are no competing interests.

Pilar Soledispa conceived and designed the experiments, performed the experiments, analyzed the data, prepared figures and/or tables, authored or reviewed drafts of the paper, and approved the final draft.

Efrén Santos-Ordóñez conceived and designed the experiments, analyzed the data, prepared figures and/or tables, authored or reviewed drafts of the paper, and approved the final draft.

Migdalia Miranda conceived and designed the experiments, analyzed the data, authored or reviewed drafts of the paper, and approved the final draft.

Ricardo Pacheco performed the experiments, analyzed the data, authored or reviewed drafts of the paper, and approved the final draft.

Yamilet Irene Gutiérrez Gaiten performed the experiments, analyzed the data, prepared figures and/or tables, authored or reviewed drafts of the paper, and approved the final draft.

Ramón Scull performed the experiments, authored or reviewed drafts of the paper, and approved the final draft.

The following information was supplied relating to field study approvals (i.e., approving body and any reference numbers):

Samples were collected for identification in the framework of the GUAY Herbarium, Facultad de Ciencias Naturales, Universidad de Guayaquil.

The following information was supplied regarding data availability:

Sequences are available in the Supplemental Files and at GenBank: MT740231, MT740232, MT740233, MT740234, MT740235, MT740236, MT740237, MT740238, MT740239, MT740240, MT740241, MT740242, MT734663, MT734664, MW300280, MW300281, MW300282, MW300277, MW300278, MW300279.

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
