# Peer review of "Molecular barcode and morphological analysis of Smilax purhampuy Ruiz, Ecuador"

_PeerJ, doi:10.7717/peerj.11028_

## Round 0.1 · original submission · Major Revisions

Please take into consideration the comments from the reviewers, and provide a detailed revised version along a rebuttal letter.

·

Basic reporting

The manuscript athough clear and unambiguous, could do with professional English proofreading throughout. The paper is technically correct text. The article conform to professional standards of courtesy and expression.

The article includes sufficient introduction and background to demonstrate how the work fits into the broader field of knowledge. Relevant prior literature have been appropriately referenced.

The structure of the article conform to an acceptable format of ‘standard sections’.

Figures are relevant to the content of the article, of sufficient resolution, and are appropriately described and labeled.

All appropriate raw data have been made available in accordance with our Data Sharing policy.

Experimental design

No research question is defined. However, it is stated how the research will fill an identified knowledge gap.

The submission clearly define the research problem, which is relevant and meaningful. The knowledge gap being investigated was identified, and statements should be made as to how the study contributes to filling that gap.

The investigation have been conducted rigorously.

Some parts of the methodology is incomplete- thus preventing repeatability/replication.

Validity of the findings

All underlying data have been provided; they are robust, statistically sound, & controlled.
The data on which the conclusions are based must be provided or made available in an acceptable discipline-specific repository.

Conclusions are well stated, linked to original research question & limited to supporting results.

Additional comments

TITLE
1. Suggested modification: Molecular barcode and morphological analysis of Smilax purhampuy Ruiz, Ecuador

ABSTRACT
2. The rhizomes was also analyzed. For what?
3. Results showed that … First report on micromorphological characteristics of the leaves, 2nd report on rhizome analysis, 3rd report on molecular barcode characterization – as per sequence of method reported above

IINTRODUCTION
4. The Introduction cannot consist of just one long paragraph with a number of different topics – each topic = separate paragraph. This reviewer attempted to create a number of self-contained paragraph, each with their own focal area.
5. The Introduction section provide NO information on Smilax purhampuy in Ecuador, which is the core species of this paper. Is it indigenous to Ecuador? What medicinal value does it have? Is it taxonomic position uncertain? etc etc.– remedy this by inserting information on S. purhampuy /or/ replace some of the general information of the genus with more specific information related to S. purhampuy.

MATERIALS AND METHODS
6. Collection of plant material - Indicate season of collection; indicate is leaves were sun-exposed leaves or shade-exposed leaves?
7. Micromorphological analysis - Incomplete: Refer to example provided in annotated manuscript

RESULTS
8. Morphological analysis - Incomplete: Refer to example provided in annotated manuscript
9. "...while other genera (indicate names of genera) were in other clades (Fig. S1)."
10. "On the other hand, the ITS2 phylogenetic tree revealed that the two Smilax samples (name them) are in a different clade (bootstrap 100).."

DISCUSSION
Microscopic analysis
11. The cross section of S. purhampuy leaf indicated that the abaxial and adaxial epidermis are uniseriate, with the existence of an easily perceptible cuticle (how thick was the cuticle?) – not mentioned in results section.
12. :In S. purampuy, a dorsiventral ....Martins and Appezzato-da-Glória (2006) to S. polyantha and Martin et al. (2013) for (which species?).
13. A peculiarity of the species studied was the fact of findingpresence of six vascular bundles of variable size at the level of the central nerve vein of the leaf (was this indicated as such in the results section?), which is in discrepancy with other Smilax species (indicate which species) that present three vascular bundles (Martin et al., 2013). Cannot have a 1 sentence paragraph
14. In other species (indicate which species) of the Smilax genus from Cuba and Guatemala, the ..."

Molecular barcode sequence
15. The genotypic characterization of plant species is important since most plants, although they belong to the same genus and species, could show considerable variation (in what?).
16. "...011, CIBE-012) were more similar between them than other Smilax species (name them)..."
17. "Furthermore, the ITS2 revealed low percentage (be specific) ..."

REFERENCES
18. Some references not found in text - check all

Reviewer 2 ·

Basic reporting

In the Introduction section, the authors focused on the medicinal value of the genus Smilax, but did not introduce the reason why DNA barcodes are needed for its identification. Perhaps it is difficult to separate it from its closely related species based on the morphological characteristics? In addition, what methods have been used for the identification of this genus? The author can elaborate on the above issues in more detail.

Experimental design

- The Smilax genus consists of 310 species, but the authors only analyzed one of them (Smilax purhampuy). What is special about this species?
- Very limited number of specimens were analyzed in this study.
- Where does the “powder drug of the rhizomes” (Line 112) come from?
- When constructing a phylogenetic tree, bootstrap replicates is generally set to 1000 to make the results more reliable.

Validity of the findings

- For the same species, it is strange that the lengths of the amplified sequences from the same marker vary greatly (Data_S1). Before the blast analysis and alignment, it is necessary to cut the sequences from the same marker with a uniform standard to avoid false positives. The sequence length of different barcodes should be mentioned in Results section. Correspondingly, the results of the phylogenetic trees also need to be updated.
- “Smilax samples” and “samples of Smilax” in Results section should clearly refer to Smilax purhampuy. Please revise thoroughly.
- After comparing all the barcodes in this study, the author should conclude which barcode is the most effective for identifying Smilax purhampuy. The discussion of candidate barcode selection for Smilax purhampuy or Smilax genus should be enriched.

Additional comments

-This work reported the micromorphological characteristics and the molecular barcodes of Smilax purhampuy collected in Ecuador. The most prominent problem lies in sample size and pre-processing of data, which need to be improved and modified.
-The English language should be further improved.

Reviewer 3 ·

Basic reporting

I believe that both the literature review on molecular methods, the description of results and the discussion of part concerning barcode analysis require improvement and extending.

1. I would advise to enrich the Introduction with an outline of the phylogenetic and taxonomic knowledge of the genus Smilax. Has any phylogenetic research been conducted on this genus? If so, it is also worth referring to them in the Discussion.

2. I think it is worth to mention that presently in GenBank database there are only few complete genomes of Smilax published, namely: four genomes representing three species. The methodology applied in the study would be also more transparent if the Authors state it clearly, that plastomes of other Smilax species which appear in The Results as blastn hits are not published as complete genomes but only chosen genes are sequenced and deposited in GB. It should be explained to the reader that the results of the blastn analysis showing similarity to a given species are based on the incomplete database.

3. The Authors should also explain why it was not possible to conduct a barcoding-gap analysis, or to use automated species identification such as Spider, SpeciesIdentifier or FASTACHAR, routinely used in studies concerning barcodes. What I mean is the fact that for these kind of analyses more than one representative for each species is needed. In this study S. purphampuy is represented by three samples collected for this study but the Genbank database unfortunately does not provide multiple representatives within species taken into comparison.

4. In my opinion, the results would definitely become more readable if one or several of the best-developed trees were presented in the main article body and not in a supplement. Instead, table 1 could be moved to supplementary information.

5. The best blastn hits described in the Results in lines 170-178 could be successfully presented in Table 2 by bolding the highest results. I think In the text The Authors should emphasize the most important results, e.g.: that the rpoC1 gene didn’t discriminate species within Smilax genus at all. I would also mention that the use of matK as a barcode didn’t discriminate S. purphampuy from S. fluminensis. The variability of rbcL sequence is also worth discussing. The identity of rbcL sequences in this study was always over 99%, however it is not known whether this was due to interspecies variation, or also largely to variation within species. The latter possibility would be indicated by the result of the ML analysis, which does not group Smilax species into species-specific clades.

Experimental design

No comment

Validity of the findings

1. In lines: 31 and 81 the Authors stated that morphological and molecular barcode analysis S. purphampuy are lacking in Ecuador. Have they been made in any other country? The reader may doubt whether the research is innovative in a given area or for the species at all.

2. Please emphasize the fact that there is little molecular data on Smilax available, what will emphasize the novelty of this work and at the same time dispel the reader's doubts about the correctness of the methodology used in the study.

3. The valuable information which I lack in Conclusion is an indication of which barcode or barcodes best identify species of the genus Smilax.

Additional comments

The manuscript entitled “Molecular barcode and morphological analysis of the medicinal plant Smilax purphampuy Ruiz collected in Ecuador” presents the results of the study on the representative of Smilaxaceae family. The studied species gained the attention of the Authors of the research because of its usage in folk medicine and because of the lack of morphological and barcode analysis for S. purphampuy in the literature.
The subject of the research is in line with the current research trends, and the problem undertaken in the research is absolutely justified. The manuscript describes the research conducted in two ways. Some of them concern morphological research, the other part molecular analysis. I do not feel like an expert in the field of morphological research, although both the methods used seem to me to be reasonably selected and presented, and the description of the obtained results is understandable, clear, consistent and exhaustive. This part gives the impression of being well thought out and thoroughly conducted. Regarding the description of research conducted with molecular methods, I have a few comments. The laboratory methods themselves are well selected and described and do not raise my concerns.
I hope that questions and detailed comments on specific chapters listed in my review will help to improve the manuscript.

Below minor comments to the manuscript are listed:
Please, standarize the nomenclature of genes – relates to the use of italics in gene names.
Line 51: „… therefore, the taxonomy classification of Smilax was confirmed” –this conclusion goes a little bit too far and is a bit of a oversimplification.
Line 133 – should be sequenced instead of sequence
Line 134 – Replace sequences with the word sequenced
Line 139 – Please, replace sequences with sequence
Line 141 – I don’t understand, what number did the Authors wanted to mention in brackets?

---

## Round 0.2 · Minor Revisions

Please take into consideration the reviewer’s comments and provide back a point-by-point rebuttal letter addressing those concerns. It is also imperative to have the manuscript edited by an English speaking reviewer or a professional copywriter.

·

Basic reporting

Minor grammar issues persists.
Minor punctuation issues persists
Minor typographical issues persists
Check maximum number of words allowed for Abstract, according to journal guidelines
Some references (Costion) (Chang), amongst other, not found in text - please check all citations in Bibliography against the text

Experimental design

Anatomical preparation not adjusted - how were the leaves handled prior to dissection? - see previous version example attached by reviewer.

Validity of the findings

Uniseriate epidermis not mentioned in results

Additional comments

Much improved manuscript.

Reviewer 3 ·

Basic reporting

The text has been thoroughly corrected according to all the reviewers’ comments. Now it is comprehensive and understandable. English language is clear and professional. In my opinion in the present form, the manuscript meets the standards of PeerJ Journal. However, there are some minor, almost cosmetic changes that must be made before publication of the manuscript. After making pending corrections in the text, I recommend the manuscript to be accepted for publication.
The most pending is spelling of Latin names in italics. I realize nowadays there is a trend to write generic names without italics, although in the Shenzhen Code from 2018 International Code of Nomenclature for algae, fungi, and plants (iapt-taxon.org) All the generic names and species names are still written in italics. Please, check the spelling in Your manuscript, also in figure captions and table headings.
The text needs to be checked for double spaces and for spaces before commas.
In line 234 there is double “the” – one has to be deleted.

Experimental design

The paper presents original study. The methods applied in the study are correct, described with sufficient details, the results were correctly interpreted and discussed.

Validity of the findings

The strong point of this study is considering both morphological and molecular studies in one paper, which unfortunately, nowadays isn't very common practice in the studies concentrated on molecular methods. The second advantage of the study is an object of a research which has been poorly investigated so far, being at the same time broadly used in traditional medicine.

Additional comments

I encourage The Authors to continue their studies on Smilax to complete the barcode sequences database and broaden possibilities of data analysis.

---

## Round 0.3 · Minor Revisions

The English is still not great. The first paragraph of the Intro is fairly representative, with lines like 'According to Cameron et al. (2006), Smilacaceae are taxonomically confused which belong to the cosmopolitan family of Liliales...'. It requires further professional proofreading.

Please check the use of the word "nerve" in the abstract. Probably you meant "vein" or "mid-rib".

---

## Round 0.4 · accepted · Accept

Thanks for addressing all the revisions and corrections requested. Now your manuscript is accepted in PeerJ.